# A COVID Dilemma: How to Manage Pregnancies in Case of Severe Respiratory Failure?

**DOI:** 10.3390/healthcare11040486

**Published:** 2023-02-07

**Authors:** Pierpaolo Di Lorenzo, Claudia Casella, Mariagrazia Marisei, Laura Sarno, Carmen Imma Aquino, Eduardo Osuna, Maurizio Guida, Massimo Niola

**Affiliations:** 1Department of Advanced Biomedical Sciences, University of Naples Federico II, 80131 Naples, Italy; 2Department of Neuroscience and Reproductive and Odontostomatological Sciences, University of Naples Federico II, 80131 Naples, Italy; 3Department of Gynecology, Obstetrics University of Piemonte Orientale, Ospedale Maggiore della Carità, 28100 Novara, Italy; 4Department of Sociosanitary Sciences, University of Murcia, 30005 Murcia, Spain

**Keywords:** COVID-19, pregnancy, ethics, information, consent, vaccines

## Abstract

To date, the impact of the COVID-19 pandemic on the world’s health, economics and politics is still heavy, and efforts to mitigate virus transmission have caused remarkable disruption. From the early onset of the pandemic, generated by SARS-CoV-2 spread, the scientific community was aware of its impact on vulnerable individuals, including pregnant women. The purpose of this paper is to highlight scientific pitfalls and ethical dilemmas emerging from management of severe respiratory distress in pregnant women in order to add evidence to this topic through an ethical debate. In the here-presented paper, three cases of severe respiratory syndrome are analyzed. No specific therapeutic protocol was available to guide physicians in a cost–benefit balance, and unequivocal conduct was not *a priori* suggested from scientific evidence. However, vaccines’ advent, viral variants lurking on the horizon and other possible pandemic challenges make it necessary to maximize the experience gained through these difficult years. Antenatal management of pregnancies complicated by COVID-19 infection with severe respiratory failure is still heterogeneous and ethical concerns must be pointed out.

## 1. Introduction

The SARS-CoV-2 virus (acronym for Severe Acute Respiratory Syndrome Coronavirus) belongs to the Coronavirus family, pathogens responsible for both common infections of the respiratory system and for severe and life-threatening forms, such as MERS (Middle Respiratory Syndrome-Eastern) and SARS (Severe Acute Respiratory Syndrome) [1]. Coronavirus Disease 2019 (COVID-19), triggered by Severe Acute Respiratory Syndrome Coronavirus-2 (SARS-CoV-2) infection, was declared a pandemic by the World Health Organization (WHO) on 11 March 2020 [2].

New mutant strains continue to appear, provoking public health shocks [3]. Since its appearance in late 2019, repercussions of SARS-CoV-2 on the world’s health, societies and economies were massive [4,5,6,7].

Data on the effects during pregnancy of previous coronaviruses (severe acute respiratory syndrome [SARS] and Middle East respiratory syndrome [MERS]) [8], along with information on other respiratory infections, such as influenza, raised concerns about the potential effects of COVID-19 during pregnancy [9].

Initially, a general tendency towards risk avoidance and lockdown masked some of the increased risks associated with SARS-CoV-2 infection in pregnancy [10,11,12].

Consequently, given the paucity of data, from the very beginning, there was a paramount effort from the scientific community to better understand the effects of SARS-CoV-2 infection on pregnancy, leading to fervent scientific activity [13,14,15].

Many studies have reported the prevalence of SARS-CoV-2 infection among pregnant women presenting to labor and delivery, with estimates ranging from 2–30% [16].

Nevertheless, these rates are difficult to compare to other populations because universal screening is not commonly performed. This situation is different in the context of pregnant women.

SARS-CoV-2 testing is often conducted as part of screening for hospital admission at delivery; however, pregnant and recently pregnant persons who test positive for SARS-CoV-2 infection are less likely to report symptoms [17].

Pregnancy is associated with increased disease severity in those infected with SARS-CoV-2 [17].

In parallel, maternal SARS-CoV-2 infection can impact pregnancy in numerous ways. The need for intensive care associated with severe disease can necessitate delivering the infant, causing an increased rate of preterm delivery. Placental infection can be associated with SARS-CoV-2 placentitis, which is related to an increased risk of stillbirth. Even in the absence of placental infection, inflammatory changes are observed in the decidua and placenta, and these may be linked to the increased risk of pre-eclampsia associated with SARS-CoV-2 infection in pregnancy. SARS-CoV-2 can also be vertically transmitted to infect the fetus, although this is uncommon [18].

The National Institutes of Health (NIH) provides COVID-19 treatment guidelines for hospitalized adults. At the time of the study, NIH guidelines recommended dexamethasone or alternate systemic steroids and remdesivir to decrease severity among hospitalized patients who require supplemental oxygen or mechanical ventilation [19,20]. There are limited data on the use of COVID-19 therapeutic agents in pregnant and lactating people. Decisions about treatment in pregnant or breastfeeding women should be made by patients and clinical teams, using a shared decision-making process that takes several factors into consideration. COVID-19 disease severity, the risk of disease progression, and the safety profile of specific medications for the fetus, infant, or pregnant or breastfeeding individuals are all factors that must be considered [20]. Recognizing the limitations of available data in pregnancy, the pregnant patient and the clinical team should consider the safety of the medication for the pregnant or lactating individual and the fetus, as well as the severity of maternal disease.

The COVID-19 Treatment Guidelines Panel recommends against withholding COVID-19 treatments from pregnant or lactating individuals specifically because of their condition [20].

Normally, the therapeutic management of pregnant patients with COVID-19 should be the same as for nonpregnant patients, with a few exceptions (AIII). Notable exceptions include the recommendation against the use of molnupiravir for the treatment of COVID-19, unless there are no other options and therapy is clearly indicated [20].

In fact, Remdesivir is the only antiviral drug that is approved by the Food and Drug Administration (FDA) for the treatment of COVID-19 but “the efficacy and safety profile of remdesivir among pregnant women with COVID-19 remain inconclusive. Even though better clinical status at baseline with earlier remdesivir treatment may result in better clinical outcomes, careful monitoring of adverse reactions and transaminase enzyme levels should be carefully monitored” [21].

When other therapies are not available, pregnant people with COVID-19, which has a high risk of progressing to severe disease, may reasonably choose molnupiravir therapy after being fully informed of the risks, particularly if they are beyond the time of embryogenesis. In most cases, the timing of delivery should be dictated by obstetric indications, rather than maternal diagnosis of COVID-19. For people who had suspected or confirmed COVID-19 early in pregnancy and who recovered, no alteration to the usual timing of delivery is indicated (i.e., >10 weeks’ gestation) [20].

The Society for Maternal-Fetal Medicine [22] supports the NIH COVID-19 treatment guidelines [20] and recommends that remdesivir and dexamethasone be offered to pregnant patients with COVID-19 who require supplemental oxygen. Nonetheless, the extent to which these guidelines are followed in hospitalized pregnant patients with COVID-19 is unknown [19].

Many clinical trials evaluating novel treatments for COVID-19 have excluded pregnant persons; however, treatment recommended for the nonpregnant population should not be withheld from pregnant patients. This includes treatment with Remdesivir, dexamethasone and monoclonal antibodies. Given that pregnancy is a risk factor for progression to serious disease, pregnant persons are eligible to receive outpatient treatment or postexposure prophylaxis with antiSARS-CoV-2 monoclonal antibodies under the Emergency Use Authorization [23].

Pregnant and breastfeeding women are excluded from participating in clinical trials during this pandemic [24]. This “protection by exclusion” of pregnant women from drug development and clinical therapeutic trials, even during pandemics, is not unprecedented [25,26].

This is another missed opportunity to obtain pregnancy-specific safety and efficacy results, because therapeutics verified for men and non-pregnant women may not be generalizable to pregnant women due to their specific condition [27]. Pregnant women should be given the opportunity to choose to be included in clinical trials for COVID-19 based on the concepts of justice, equity, autonomy and informed consent. Even during the Ebola virus epidemic, pregnant women were excluded from all therapeutic and vaccine-development trials. This automatic disqualification denies pregnant women the potential for benefit given to other patients [28].

## 2. Materials and Methods

In our tertiary Italian Centre, almost 350 pregnant women with COVID-19 infection have been managed since March 2020. Here, we present three interesting cases.

### 2.1. Case 1

#### 2.1.1. Clinical History

A 31-year-old Caucasian woman was admitted at the Emergency Department of University Hospital Federico II of Naples at 27 weeks of gestation, due to worsening of dyspnea.

Her medical history was silent; she had neither comorbidities nor surgical history, except for the three CSs (Cesarean Section). The body mass index (BMI) on admission to the prenatal care Covid Unit was 25.9 kg/m^2^, with length 165 centimeters (cm) and weight 70 kilograms (kg). Her pregnancy had been without complications.

She reported a positive SARS-CoV-2 nasopharyngeal swab, which she performed after the onset of fever and loss of appetite. Two days after the diagnosis, due to the onset of dyspnea, she started a home therapy with deltacortene and amoxicillin–clavulanic acid.

The woman was not vaccinated, nor were her whole family, most of whom resulted equally positive for SARS-CoV-2.

At admission, she was febrile, with an oxygen saturation (SpO_2_) of 82% on room air, respiratory rate of 40–50, blood pressure 110/70 mmHg, pulse 90 beats/minute and temperature 38 degrees Celsius (°C).

#### 2.1.2. Therapeutic Management

The patient was given dexamethasone 6 mg, paracetamol for pain and fever relief and ceftriaxone 2 g; she also received thromboprophylaxis with enoxaparin 4000 units/day subcutaneously. Her SpO_2_ remained in the 80s by pulse oximetry, despite 8 L/min of oxygen by Venturi mask, and her ABG (Arterial Blood Gas test) showed a pH 7.4, pCO_2_ 31.9, PaO_2_ 72.4, HCO_3_–std 22.2, Base deficit −2.7 and O_2_ saturation 90%. Chest X-ray demonstrated diffuse bilateral consolidations; pulmonary ultrasound showed loss of normal echostructure of the parenchyma due to the presence of diffuse B lines; and a clinical exam revealed coarse and diminished breath sounds. Obstetric examination including an abdominal ultrasound showed no abnormalities.

For two days, the patient’s clinical condition was closely monitored and seemed relatively stable. Antenatal corticosteroids prophylaxis was carried out with two doses of 12 mg betamethasone every 24 h. Despite ongoing antibiotic therapy, the patient had an increase in white blood cells (18.520/uL) and PCR (107.10 mg/L), then ceftriaxone was discontinued and meropenem and azithromycin were added to the therapy.

Three days after the hospitalization, the patient had a drastic worsening of respiratory function; she was agitated and not cooperating.

Several C-PAP attempts had been unsuccessfully made due to the lack of compliance of the patient to the Boussignac mask and to the Helmet. Because of the persistent worsening of the patient’s clinical and respiratory condition, the woman was intubated and transferred to the Intensive Care Unit. Her obstetric condition was stable. A combined evaluation between obstetric anesthesiologists and neonatologists before intubation had been necessary, and an appropriated counselling had been carried out to the mother.

Due to the critical clinical condition of the patient, a fetal neuroprotection therapy with Magnesium Sulfate was given.

In the Intensive Care Unit, the patient underwent daily hematochemical analyses and ABG. Several adjustments in the ventilation mode had been made to ensure the best oxygenation.

Her condition deteriorated on day 7 of the admission.

Gynecologists and virologists were immediately alerted to decide on the management of the patient. Despite her worsening condition, the fetal situation was stable and normal intermittent US controls were recorded. Since the general condition of the patient had worsened, a decision was made to deliver by CS (27 + 5 gestational weeks), on maternal indication.

#### 2.1.3. Pregnancy Outcomes

A male baby was delivered, with birth weight 1200 g, birth length 37 cm and head circumference 24.5 cm. The cord was clamped immediately after birth. The Apgar score was 5 and 7 at 1′ and 5′, respectively.

Nasopharyngeal swabs for SARS-CoV-2 detection were collected at 48 and 96 h of life and were found to be negative in both instances.

The baby is still alive, but in severe clinical condition due to severe prematurity.

### 2.2. Case 2

#### 2.2.1. Clinical History

A 31-year-old Caucasian woman was at 27 weeks of gestation. Due to the onset of dyspnea, a SARS-CoV-2 nasopharyngeal swab was performed and it resulted positive. The patient was transferred to the nearest local COVID Center. The woman was not vaccinated against SARS-CoV-2.

#### 2.2.2. Therapeutic Management

At the time of admission, she was febrile with an oxygen saturation (SpO_2_) of 87% on room air and a respiratory rate of 35–45, blood pressure 100/65 mmHg, pulse 85 beats/min and temperature 39.3 degrees Celsius (°C). The patient was given dexamethasone 4 mg (twice a day), paracetamol for fever and pain relief and amoxicillin—clavulanic acid 1 g (three times a day). She also received thromboprophylaxis with Enoxaparin 4000 units/day subcutaneously. Obstetric examination including an abdominal ultrasound showed no abnormalities.

Due to severe respiratory failure, an urgent CT lung and abdomen scan was performed. The CT scan demonstrated multiple bilateral pulmonary consolidations affecting all the lobes, but mainly extended to the level of the lower ones, no pleural effusion and a minimum pericardial effusion distribution (total severity score 19/20). The clinical exam revealed coarse and diminished breath sounds. 

For critical clinical and instrumental conditions, it was decided to start NIV with Helmet (ΔP 15 mmHg, PEEP 7.5 cm H_2_O, FiO_2_ 60%). Her ABG showed a pH 7.42, pCO_2_ 31, PaO_2_ 55, HCO_3_–std 22.0, Base deficit −4.4 and O_2_ saturation 89%. 

At the same time, the transfer to a hospital equipped with a COVID—Intensive Care Unit and Neonatal Intensive Care Unit (NICU) was requested.

After 24 h in NIV, a worsening of respiratory function was manifested by clinically evident fatigue. After mild sedation, orotracheal intubation was performed. 

On the second day of hospitalization, the patient was transferred to the prenatal care Covid Unit of Federico II University Hospital in Naples in severe ventilatory condition, despite orotracheal intubation. The fetal situation was stable and US controls were recorded. Since the general condition of the patient had worsened, a decision was made to deliver by CS (26 + 5 gestational weeks), on maternal indication. Prior to CS, only one dose of betamethasone 12 mg was taken for antenatal corticosteroids prophylaxis. 

The patient died due to multiorgan failure almost two months after delivery.

#### 2.2.3. Pregnancy Outcome

At 48 h after CS, the patient had fever (38–40 °C) and tachycardia (120 bpm). The respiratory conditions remained extremely serious. On the sixth day of hospitalization, serum galactomannan and coproculture for Clostridium Difficile serum tested positive. 

From the respiratory point of view, the patient’s condition had been stationary, but serious for weeks. The mechanical ventilation mode (IPPV) was never changed and a total of seven pronation cycles were carried out. The newborn died due to severe prematurity after 19 days.

### 2.3. Case 3

A woman pregnant at 26 gestational weeks referred to Federico II University Hospital in Naples for fever, cough and dyspnea. She was carrying a dichorionic diamniotic (DCDA) twin pregnancy complicated by gestational diabetes, with healthy fetuses. She was not vaccinated against SARS-CoV-2.

#### 2.3.1. Clinical History

She tested positive for the Covid infection. After two days, she was transferred to the Infectious Diseases Department. Diagnostic imaging revealed widespread bilateral interstitial pneumonia, complicated by gross inflammatory thickening in the left mid-basal site (due to possible bacterial superinfection). During hospitalization, the patient had a rapid worsening of clinical conditions with a desaturation, despite high-flow oxygen therapy, for which it was decided to transfer to intensive care.

#### 2.3.2. Therapeutic Management

She underwent non-invasive ventilation with a helmet. Despite the established therapy, the patient was tachypneic and maladapted, with evident muscle fatigue, for which she was intubated, and subjected to mechanical ventilation. The patient was given Enoxaparin 4000 UI twice a day, dexamethasone 12 mg, paracetamol, iron and folic supplementation, Cefepime 2 g twice a day, Azithromycin 500 mg, Clindamycin 600 mg, Meropenem 2 g, Fluconazole 200 mg, casirivimab, and imdevimab monoclonal antibodies, KCL retard 600 mg, N-acetylcysteine mucolytic 300 mg. 

From repeated obstetrical ultrasound checks, the pregnancy resulted in normal evolution. The patient’s precarious conditions with continuous phenomena of desaturation made urgent a CS at 27 weeks of gestation, performed after fetal neuroprophylaxis with magnesium sulfate. Her last ABG showed: pH 7.28, pCO_2_, 80.8, PaO_2_ 28.3, HCO_3_–std 31.5, HCO_3_–act: 37.8, Base excess 8.8 and O_2_ saturation 47.5%, Hct: 3.6, glycemia: 123 mg mg/dL, Hb 10.9, Calcium: 4.6 mg/dL, Cl: 92 mmol/L, Na: 135 mmol/L and K: 3.29 mmol/L.

In the following days, the patient’s clinical situation continued to worsen, with diffuse pneumothorax to the right, flap and pneumomediastinum. Such a framework would lead to cardiorespiratory failure and the patient’s death a month after the cesarean section.

#### 2.3.3. Pregnancy Outcome

Newborns (a boy and a girl) were transferred to the Neonatal Intensive Care Unit. At birth, their Apgar scores were 4-7-7 for him and 1-5-7 for her at 1′, 5′, and 10′, respectively. The newborns’ birth weight was 960 g for the male and 730 g for the female. Positivity to SARS-CoV-2 was ascertained for both of them. They suffered from necrotizing enterocolitis and were transferred to a tertiary care center and then discharged home after the appropriate treatments and the attention given in case of prematurity.

## 3. Results

In these cases, what seems to be clear was the absence of a definite obstetric therapeutic protocol. Patients were managed with a multidisciplinary approach. The fetal situation was stable. All efforts converged to save the mother by actuating all different therapeutic options, including pronation, but the maternal survival rate was already compromised. All cases converge to the result that the best clinical treatment under specific circumstances was given.

## 4. Discussion

Through this pandemic period, that cannot be considered to be over yet, at least three serious issues were raised in this peculiar care setting. 

Firstly, considering the possibility of intubation and transfer to the Intensive Care Unit, should the patients receive counselling for termination of pregnancy (TOP) or is it better to “wait and see”? 

This apparently pointless consideration came from the lack of evidence regarding possible effects of prenatal exposure to high doses of oxygen during fetal life, or to some drugs, especially in the second trimester of pregnancy [29]. In fact, when necessary, it has been recommended to target PaO_2_ at 70 mmHg, with oxygen saturation between 94% and 98% [30].

From an infectious disease perspective, delivery can give the patient the chance to be treated with antivirals, of which the safety in pregnancy is uncertain. An example of this therapy is Remdesivir. It is an antiviral drug approved for COVID-19 treatment in the early stages of the pandemic, but data on pregnancy are incomplete [31,32]. Despite the lack of real-life data, remdesivir registration trials showed that among 1092 enrolled patients who received it, it had a median recovery time of 10 days, as compared with 15 days among those who received placebo (rate ratio for recovery, 1.29; 95% CI, 1.12 to 1.49; *p* < 0.001, by a log-rank test) [33]. To date, data about the use of this drug in these clinical conditions are not yet definitive [21].

Literature is still poor in cases of use of some other antiviral drugs, such as tocilizumab, or monoclonal antibodies in pregnant women, despite significative results in terms of severe morbidity [34]. 

The timing and the mode of delivery are to be chosen considering gestational age and maternal symptoms; no clear data are available in case of critical maternal condition during the different gestational periods [35]). According to NIH COVID-19 treatment guidelines, the use of specific treatments should not be avoided because of theoretical concerns related to the safety of therapeutic agents in pregnancy [36]. Therefore, these drugs should not be used as first-line therapy, and it is not established whether the patient should be informed of this risk and decide to proceed for eventual TOP before the intubation. 

Secondarily, it is not known whether a TOP, as last therapeutic chance, could improve maternal outcome and change the natural history of the disease when maternal COVID-19 is severe [37]. TOP might be used when there is difficulty in ventilating the mother. This would usually be due to raised intra-abdominal pressure from the pregnant uterus, and this would more likely be in the third trimester (or earlier, in the case of, e.g., twins) or with evidence of fetal brain damage secondary to severe maternal hypoxia. In light of this, TOP should be expressly considered only in serious conditions, beyond any maternal will. Unfortunately, most of the reported cases presented mild respiratory symptoms and are related to the third trimester of pregnancy, whereas there is a lack of evidence related to mid-trimester [38]. 

Thirdly, what is the best method of induction in the case of TOP? In this case, there is no evidence to state a traditional protocol that could not worsen maternal conditions. Considering that prostaglandins are contraindicated in the case of severe asthma, mechanical methods could be preferred in these patients. 

If at the very beginning of pandemics, with the unknown effect on pregnancies and the lack of therapeutic safe treatments, the biggest dilemma was how to treat a pregnant woman affected by COVID-19, with the advent of vaccines the dilemma has shifted to the healthy pregnant [39]. In both cases, only informed consent can really realize a positive connection and a real cooperation between doctors and obstetricians in the sense of protection.

Data on vaccine coverage suggest that pregnant women are less likely to receive a COVID-19 vaccine, despite their increased risk for severe disease and the risks of adverse pregnancy and neonatal outcomes if infected. A percentage coverage of 71% [40] suggests that there is still informative work to do.

More recent meta-analysis of 111 studies, which compared outcomes for pregnant patients infected with SARS-CoV-2 with those who were not infected, found that infection significantly increased the odds of premature delivery (OR 1.48, 95% CI 1.22–1.8), pre-eclampsia (OR 1.6, CI 1.2–2.1), stillbirth (OR 2.36, CI 1.24–4.46), neonatal mortality (OR 3.35, CI 1.07–10.5) and maternal mortality (OR 3.08, CI 1.5–6.3) [41].

In Scotland, 68% of the pregnant population was unvaccinated by October 2021, but unvaccinated individuals accounted for 77.4% of all SARS-CoV-2 infections, 90.9% of COVID-19 hospitalizations and 98% of intensive care unit admissions among pregnant people; furthermore, all perinatal deaths following SARS-CoV-2 infection in pregnancy occurred in unvaccinated individuals [42].

Most experts believe that SARS-CoV-2 is likely to become endemic. Thus, the continued collection of data on the effects of SARS-CoV-2 infection during pregnancy and the effects of COVID-19 vaccines are needed [43].

In the presented cases, a multidisciplinary approach based on informed consent [44] could have been the keystone of the whole pregnancy management.

## 5. Conclusions

There is a need for evidence about pregnancies complicated by COVID-19 with severe respiratory complications.

More studies could be helpful in order to answer a new ethical dilemma: can TOP be considered as an option [45] in case of severe COVID-19 infection from the second trimester of pregnancy or in case of any other severe viral infections?

Apart from a specific dilemma, one fundamental instrument should be taken into account as being the milestone of the care relationship: informed consent.

It is the legitimation of the medical act. A truly informed consent is based on adequate information and is the trademark of cooperation between doctor and patient, a real therapeutic alliance [46].

It was clear during these years that the scientific world did not have immediate knowledge about COVID-19 pathogenesis, risk of maternal infection and treatment during pregnancy. Considering these limitations, all care workers made a great effort to treat population, with special regards to vulnerable individuals [47].

Dilemma is always a matter of value. Even in the complete darkness of a new situation to confront with, only a real informed consent guarantees medical doctors and health professionals, in general, a correct way to act and to avoid future malpractice claims [48].

Given the increasing connectedness of the world’s population, climate change, and the increasing encroachment of human populations on wildlife habitats, the emergence of another infection with global effects is likely. Knowledge derived from the COVID-19 pandemic can be fruitful in response to emerging infections in the future [49,50].

In conclusion, a patient-centered multidisciplinary approach made it possible to overcome pandemic challenges and seems to be crucial in relation to future scenarios.

## Data Availability

Not applicable.

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
