# Peer review of "A COVID Dilemma: How to Manage Pregnancies in Case of Severe Respiratory Failure?"

_healthcare, 2023, doi:10.3390/healthcare11040486_

Round 1

Reviewer 1 Report

The current manuscipt is short of details for the three cases studies and therefore should be improvded significantly in order to convince the readers about the paper's validity and soundness.

Author Response

We thank the Revisor for his precious opinion. We tried to offer a clear and complete presentation of the three clinical cases, knowing that an ethical perspective was the real core of the article. We will continue to work on our way to present clinical cases in order to be able to give more details while describing.

Reviewer 2 Report

This is clearly an important topic. As noted in the checklist above, this article needs extensive editing of English language and style that currently interferes with the readability of the article. One sentence paragraphs should be avoided.

The literature review at the beginning of the article gives a good sense of of the impact of COVID on pregnant women and the various treatments available. The authors make the important point that pregnant women with COVID should be able to participate in clinical trials.

The authors make the important point that no therapeutic protocol was available to guide physicians in their treatment of pregnant women with respiratory illness. Should physicians be guided by only a cost-benefit analysis? What about concern for the life of the patient and child?

350 pregnant women were treated for COVID in the Italian Centre. Why were these 3 cases chosen for review? Additional explanation is needed, other than these cases were deemed "interesting." Was vaccination status a criterion? That a caesarean section was needed? The fact that a mother died? That a child died?  Some of the information included about the treatment the mother received is not really accessible to those of us who are not physicians. The 3 cases are described but analysis is lacking. The Results section is poor. What information did the authors glean from these case reviews? Did they conclude that the pregnant women received the best care under the circumstances, considering the lack of a protocol?

The strength of the article lies in the 3 issues described in the Discussion section. Clearly it is critical to determine whether "wait and see" is preferable to termination of pregnancy. Second, whether termination of pregnancy can improve the outcome for women. The third point about the best method of induction in a case of termination of pregnancy is not clear and needs to be more explicit. What did the authors learn that can contribute to the development of a protocol? Can the authors offer recommendations for what a developing protocol should contain? Can the authors take the lead and offer something that could serve as a first draft of a protocol? This could be placed as a figure in the article and then would allow readers to respond and contribute to subsequent drafts.

The importance of this article lies in these three points in the Discussion section. It is for this reason that despite the problems with the article, I hope the authors will continue to work on it and resubmit.

Author Response

1)

We would like to thank the Revisor for the unvaluable comments sent and the important questions made. Our manuscript is centered on an ethical dilemma. Our priority is always to care mother and newborn’ s Life.

We hope that this effort could guide every clinician, but no univocal answer or single point of view can be presented. With the evolution of pandemic and the relative conprehension, our future objective is to give a chance to both mother and child, without the need for a cost-benefit analysis.

2)

We choose to recruit three emblematic cases in terms of disease evolution and clinical condition for both mother and child. The absolute interest of the three cases could be linked, in our opinion, to the worst scenario in Obstetric field: the death or severe disease of mother/child. Our intent was to make clear that not all therapeutic “weapons” were available at that time with success.

3)

Our article was based on the evolution of the three cases rather than on a point by point clinical case description. We preferred not to indulge in many clinical details and to focus only on the main data.

4)

We enlarged the results section, citing the precious Revisor’s words: in all presented cases was given the best clinical treatment under the specific circumstances (line 274-275).

5)

This is for sure an excellent hint. Scientific world is surely working on pregnancy management and Covid disease. Nevertheless, based on the information given at the time of the three cases we could imagine a protocol for next publication, but it is not the aim of this work. We would like to study further and try to suggest in future, as new scientific achievements come up, more than some sort of a protocol (surely unavailable at the time of the three clinical cases) that could be also an “ethical help” under different medical circumstances related yo the topic.

The article has been revised by a native English speaker.

Author Response

All Authours have read the comments enthusiastically. We really appreciated your kind words and your suggestions on our piece of work.
In fact, we have added the refernce you suggested at line 39 and read it with big interest.

We therefore would like to thank the Revisor, once again, for understanding the central meaning of the manuscript, making it possible to contribuite to a peculiar and paramount topic.

Round 2

Reviewer 1 Report

The authors' revised manuscript is much improved, and I would suggest that the authors cite the following references as they are very relevant to the research question and could increase the paper's research impacts:

1. Bao, Zhengyang, and Difang Huang. "Shadow banking in a crisis: Evidence from FinTech during COVID-19." Journal of Financial and Quantitative Analysis 56, no. 7 (2021): 2320-2355.

2. Baqaee, David, and Emmanuel Farhi. "Keynesian production networks and the covid-19 crisis: A simple benchmark." In AEA Papers and Proceedings, vol. 111, pp. 272-76. 2021.

3. Li, Nan, Muzi Chen, and Difang Huang. "How Do Logistics Disruptions Affect Rural Households? Evidence from COVID-19 in China." Sustainability 15, no. 1 (2022): 465.

4. Wouters, Olivier J., Kenneth C. Shadlen, Maximilian Salcher-Konrad, Andrew J. Pollard, Heidi J. Larson, Yot Teerawattananon, and Mark Jit. "Challenges in ensuring global access to COVID-19 vaccines: production, affordability, allocation, and deployment." The Lancet 397, no. 10278 (2021): 1023-1034.

Author Response

Thank you for your kind reply to our efforts.

We would also like to thank you for the bibliographic contributions that we have read with interest. We have decided to include all of them in our manuscript and references for the perfect adherence to the topic.